# Events Detection of Anticipatory Postural Adjustments through a Wearable Accelerometer Sensor Is Comparable to That Measured by the Force Platform in Subjects with Parkinson’s Disease

**DOI:** 10.3390/s22072668

**Published:** 2022-03-30

**Authors:** Tiziana Lencioni, Mario Meloni, Thomas Bowman, Alberto Marzegan, Antonio Caronni, Ilaria Carpinella, Anna Castagna, Valerio Gower, Maurizio Ferrarin, Elisa Pelosin

**Affiliations:** 1IRCCS Fondazione Don Carlo Gnocchi, 20148 Milan, Italy; tlencioni@dongnocchi.it (T.L.); mmeloni@dongnocchi.it (M.M.); tbowman@dongnocchi.it (T.B.); amarzegan@dongnocchi.it (A.M.); icarpinella@dongnocchi.it (I.C.); acastagna@dongnocchi.it (A.C.); vgower@dongnocchi.it (V.G.); 2The Biorobotics Institute, Scuola Superiore Sant’Anna, 56121 Pisa, Italy; 3IRCCS Istituto Auxologico Italiano, Department of Neurorehabilitation Sciences, Ospedale San Luca, 20122 Milan, Italy; antonio.caronni@gmail.com; 4Department of Neuroscience, Rehabilitation, Ophthalmology, Genetics and Maternal Child Health, University of Genoa, 16132 Genoa, Italy; elisa.pelosin@gmail.com; 5IRCCS Ospedale Policlinico San Martino, 16132 Genoa, Italy

**Keywords:** anticipatory postural adjustments, inertial measurement unit, Parkinson’s disease, home-based rehabilitation, gait initiation

## Abstract

Out-of-the-lab instrumented gait testing focuses on steady-state gait and usually does not include gait initiation (GI) measures. GI involves Anticipatory Postural Adjustments (APAs), which propel the center of mass (COM) forward and laterally before the first step. These movements are impaired in persons with Parkinson’s disease (PD), contributing to their pathological gait. The use of a simple GI testing system, outside the lab, would allow improving gait rehabilitation of PD patients. Here, we evaluated the metrological quality of using a single inertial measurement unit for APA detection as compared with the use of a gold-standard system, i.e., the force platforms. Twenty-five PD and eight elderly subjects (ELD) were asked to initiate gait in response to auditory stimuli while wearing an IMU on the trunk. Temporal parameters (APA-Onset, Time-to-Toe-Off, Time-to-Heel-Strike, APA-Duration, Swing-Duration) extracted from the accelerometric data and force platforms were significantly correlated (mean(SD), r: 0.99(0.01), slope: 0.97(0.02)) showing a good level of agreement (LOA [s]: 0.04(0.01), CV [%]: 2.9(1.7)). PD showed longer APA-Duration compared to ELD ([s] 0.81(0.17) vs. 0.59(0.09) *p* < 0.01). APA parameters showed moderate correlation with the MDS-UPDRS Rigidity, Characterizing-FOG questionnaire and FAB-2 planning. The single IMU-based reconstruction algorithm was effective in measuring APAs timings in PD. The current work sets the stage for future developments of tele-rehabilitation and home-based exercises.

## 1. Introduction

Gait initiation (GI) is a complex transient task performed between the quiet standing posture and steady state walking, requiring a shift from a static stable state to a, relatively less stable, dynamic state of movement [1]. For this reason, it is a challenging task which demands balance and postural control due to a decreasing base of support from a two-leg stance to an alternating single leg stance. Prior to step initiation, anticipatory postural adjustments (APAs) act to accelerate the center of body mass forward and laterally over the stance leg by moving the COP posteriorly and toward the swing leg [2]. To capture the complex physiological changes that occur during the transition from a quiet standing to the first step, following a stimulus to move (e.g., “go”), GI has been divided into three phases [3]. The first phase, related to the motor planning, begins with the “go” stimulus and ends when the APA begins. The second phase, related to the postural adjustments, extends from the beginning to the end of APA, and aims to shift the body weight towards the stance leg for stabilizing the body to prevent falls when gait begins. The third phase, related to the gait execution, extends from the end of APA to the completion of the first step. 

APAs have been extensively studied in persons with Parkinson’s disease (PD), because of their balance deficits, gait impairments and frequent falls [4]. PD is characterized by motor signs resulting from a degenerative loss of dopaminergic neurons in the substantia nigra as well as in multiple motor and non-motor regions of basal ganglia [5]. The basal ganglia are understood to be essential in planning and initiating movement, and APAs are consequently abnormal in PD [2,6]. Several studies have shown that subjects with PD have a reduced magnitude of muscle activity and abnormal muscles co-contractions during the postural adjustments with subsequent alteration in postural balance [2,7,8]. This results in prolonged timing of the GI, decreased propulsive forces and reduced COP displacements, compromising the shift of the body mass over the stance limb. Impaired APAs are associated with gait start hesitation or gait freezing that leads to falls, injuries, and fear of falling with a substantial worsening of several activities of daily living [9]. For this reason, the identification of a behavioral measurement that can describe balance disturbances in PD is highly needed, especially in a home-based unconfined context. This would allow monitoring the occurrences of dangerous situations and defining a personalized training program to improve APAs. This last aspect is particularly important in light of the results of a recent study that found a preserved flexibility of dynamic postural control in individuals with PD [10]. Indeed, home-based training, applying IMU-based systems, has been recently developed to improve gait performance, gait-related activity and health-related quality of life in people with PD, and results are encouraging [11,12,13]. However, in these studies, gait training was focused on the execution phase only, without considering the planning phase and postural adjustment of the APA, which are fundamental for a correct gait execution. Moreover, it is worth considering that recently, IMUs have been used for evaluating GI and the APA with a fair success [14,15]. Nevertheless, for the automatic detection of the APA onset, the toe-off and the heel-strike, the proposed protocols require either a minimum of three IMUs [16] (placed on the trunk and on the shanks) or the integration of data coming from footswitches [17] or pressure insoles [18]. The need for multiple sensors located on the body restricts the usability of these protocols in patients with neurological disease [19] both in home-based and clinical settings [20,21,22,23,24]. Furthermore, these approaches are limited because they require the subject to stand in a pre-specified manner, and often require a relatively extensive calibration phase. 

To address this gap, Gazit et al. [3] developed an algorithm for the detection of GI events from a single wearable sensor. The published algorithm allows evaluating the above reported three phases of GI and identifying which mobility impairments they are related to: planning, postural adjustments, or execution. The use of this algorithm in the clinical practice could help the development of personalized physiotherapy training for gait impairments. APAs-focused training has already been shown to be effective in the elderly, improving their postural control, functional balance, mobility, and quality of life [25]. Indeed, elderly subjects enhanced their anticipatory postural action and achieved a greater body stability, resulting in better performances also in untrained tasks. In the light of these results, it is desirable to introduce APAs-focused exercises in PD patients given their impairment in APAs.

In consideration of the limitation of using the multiple wearable sensors approaches reported above, the twofold aim of this work was to evaluate: (1) if the acceleration-based algorithm described in [3], that needs a single sensor, has a metrological quality comparable to that of the gold standard for APA detection (i.e., force platforms) in PD, and (2) its concurrent validity with PD disease-specific clinical scale. The potential validity of a simple GI testing system, made up of a single sensor placed on the lower back, would allow home-based monitoring and could contribute to improve APA rehabilitation treatments in PD patients.

## 2. Materials and Methods

### 2.1. Participants

Twenty-five subjects with mild-to-moderate idiopathic PD (thirteen without and twelve with previous experiences of FOG) [26] and eight healthy controls (ELD) participated to the study, after giving their informed consent. Subjects with PD that met the following inclusion criteria were recruited at the IRCCS Fondazione Don Carlo Gnocchi in Milan: age > 18 years, diagnosis of idiopathic PD (according to the United Kingdom Parkinson’s Disease Society Brain Bank criteria), Hoehn & Yahr between 2 and 3, Mini Mental State Examination ≥24, and able to walk unassisted. Exclusion criteria were: deep brain stimulation implant; history of neurologic disorders (except PD); visual, orthopedic, or vestibular impairments that could hamper task performance; and need of hearing aids. Subjects with PD were tested in their practical ON-medication state (approximately 1 h after taking their antiparkinsonian medications).

Considering the main aim of the present study (i.e., validation of APA measures from a single accelerometer sensor), the sample size calculation was estimated using previous published data related to the Bland–Altman mean difference (GRF vs. IMUs) of the APA duration parameters. Bonora et al. [15] found a bias of −0.02 s with a SD of 0.13 s—hence, estimating a maximum allowed difference comparable to the published LOA, a minimum sample size of 23 is recommended for the Bland–Altman method with α  =  0.05 and power (1 − β)  =  0.9 [27].

### 2.2. Clinical Assessment

Disease severity was evaluated with the section III of the MDS–Unified Parkinson Disease Rating Scale [28]. The patients’ more affected side was determined on the basis of MDS-UPDRS III items in which a score for each side is available (i.e., items from 20 to 26): the right and left symptom scores were calculated and the more affected side was individuated as the one with the higher score. Subsequently, the scores for tremor, rigidity of extremities, and leg agility were calculated for the most impaired side [29].

The clinical assessment included the administration of the following clinical scales: the short physical performance battery (SPPB range from 0, severe impairment, to 12, normal [30]), the Four Step Square Test (FSST, lower values-better performance [31]), and the Modified Dynamic Gait Index for the dynamic balance (MDGI, range from 0, severe impairment, to 64, normal [32]) for balance and motor skills; the Frontal Assessment Battery and its three subscales for the executive functions (FAB range from 0, severe impairment, to 18, normal, FAB-1 linguistically mediated EF, FAB-2 planning, and FAB-3 inhibition [33]); and, finally, the Characterizing Freezing of Gait Questionnaire for the freezing of gait episodes experience (C-FOG, 0 indicate no FOG [34]).

### 2.3. Procedures

Participants stood on the first force platform with their feet in parallel at hip-width and were asked to initiate a gait in response to auditory stimuli provided through a headphones set and to stop as soon as possible with both feet on the second force platform. The protocol included five trials and the auditory stimulus was a neutral voice saying ‘go’. No indication was given on the starting foot. 

The wearable measurement system used consists of a single IMU with a 16-bits resolution, analog-to-digital converter and a full scale of ±16 g (Cometa, Italy, 25 March 2022, https://www.cometasystems.com/), equipped with a MEMS sensor (25 March 2022, https://invensense.tdk.com/products/motion-tracking/9-axis/mpu-9250/). The device has a low weight (5.3 g) and small size (32 mm × 24 mm × 7 mm), is waterproof, and is equipped with a wireless interface for real-time data streaming, as well as 1 GB of on-board memory which allows to store locally up to 6 h of continuous measurements. For the study purposes, only data of the triaxial accelerometer were analyzed at a sampling rate of 140 Hz with a 16-bits resolution. The sensor was fixed with an elastic band on the trunk at L5 level. The acceleration signal was processed using a 4th-order, band pass Butterworth filter between 0.2 and 4.5 Hz.

Ground reaction forces and center of pressure (CoP) displacement were measured with the force platforms, considered as gold standard, at 1000 Hz (BTS, Milan, Italy). Retroreflective markers were placed on malleolar anatomical landmarks (BTS, Italy). COP displacements recorded from the force platforms were filtered with a fourth-order, zero-lag, low-pass Butterworth filter with a cut-off frequency of 10 Hz.

A custom-made Software, developed in Visual Studio dot net (Microsoft, Redmond, WA, USA) environment, recorded the accelerometric data and controlled the trial onset and offset, the auditory stimulus presentation and the synchronization with the optoelectronic system including the GRFs. For each stimulus sent to the headphones, a synchronous trigger signal was sent to SMART Capture software (BTS, Italy).

APAs timings quantification were calculated from three automatically-detected time points: (1) APA onset, (2) toe-off, and (3) heel-strike both from the CoP data [2] and the trunk acceleration [3]. 

Briefly, the identification of the time points from the force platforms and optoelectronic signals was calculated as follows [2]: (1) the instant of APA onset was identified as the first frame in which both antero-posterior and medio-lateral components of the CoP velocity were negative, (2) the toe-off of the swing limb (TO_swl_, Figure 1A) was identified as the frame in which the position of CoP attained the maximum distance from the line identified by the position of CoP at APA onset and at the toe-off of the stance leg (i.e., the last frame on the GRF signal), and (3) the heel-strike of the swing limb (HS_swl_) were calculated as the frame in which the malleolar marker antero-posterior velocity reached the zero value. 

The same temporal instants were then extracted from the accelerometer data of the wearable inertial system (Figure 1B) as detailed in [3] and briefly reported below. The APA onset calculation was performed separately for each of the three axes of the accelerometer signal through the two steps approach reported in the following. The overall APA onset instant was determined as the earliest detected APA time-point of the three axes. For each axis, the APA calculation consisted in the following steps: (Step 1) the instant (APA_change_) were identified, within a temporal window from 0.5 s prior to the stimulus until 1.2 s after it, as the time frame at which the sum of the residual error was minimal, indicating a significant change of the signal mean and slope. (Step 2) From the APA_change_ the algorithm searched backwards to the first local minimum, which was identified as the instant of APA_onset_. The TO_swl_ was identified from the vertical accelerometer data in the temporal window from 800 ms after the stimulus until 2 s after it, through the following four steps. In the step 1, the algorithm identified the signal positive peaks with amplitude over the threshold set to 1% of the values between 10th and 90th percentile of the signal (vector Peak_va_). In the step 2, the peak with maximum amplitude over the threshold set at 1/4 from percentile 95th of the values (Peak_va_max_) was selected among the Peak_va_. Then, during the (step 3), in the temporal window from 1.5 s until 0.5 before the frame of Peak_va_max_, the instant of the first step (I_Peak_FS_) was calculated as the signal peak (Peak_FS_) closest to the temporal window closure. Finally (step 4), the TO_swl_ was defined as the zero-crossing point in the vertical acceleration just before the I_Peak_FS._ The HS_swl_ is detected in the vertical acceleration axis prior to I_Peak_FS_ as the point where the signal crossed the 20% value of the Peak_FS._ A schematic summary of the main steps of the algorithm for the APA instants identification is reported in Table 1.

Based on the timing of APA_onset,_ TO_swl_, and HS_swl_, the following GI measures were computed for the trials of the participants (Figure 2A) [1]:

Time-to-APA, time from the “go” to the beginning of the APA (i.e., APA Onset);Time-to-toe-off, time from the “go” to the APA end, calculated as the toe-off event of the swing leg;Time-to-heel-strike, the time from the “go” to the heel-strike of the swing leg;APA duration, the time from the beginning (i.e., APA onset) to the end (i.e., Toe-off) of the APA waveform;Swing phase duration, the time from the toe-off to the heel-strike of the swing leg.

### 2.4. Statistical Analyses

The data processing and the statistical analysis were performed using Matlab (MathWorks, Natick, MA, USA) and SPSS (IBM, Armonk, NY, USA). For each subject, variables were averaged over the five trials. The agreement in measures between the force platforms and IMU was investigated through the Bland–Altman analyses and the intra-class coefficient correlation (ICC). For the former analysis, bias or systematic error, coefficient of variation (CV), and lower and upper limits of agreement (LOAs) were calculated. For the latter, an alpha model, and two-way mixed and absolute agreement were adopted [35]. The ICC was interpreted by the Fleiss’ classification using the following thresholds: below 0.40 indicated poor reliability; between 0.40 and 0.75, fair to good reliability; and above 0.75, excellent reliability. Mean absolute errors (MAEs) between instants recognized from force platforms and IMU were averaged among all subjects. Age, anthropometric data, and APA parameters were compared between PD and ELD using parametric tests (unpaired *t* test) since the normality of the data distribution was satisfied (Shapiro–Wilk’s method). Between-group (PD/ELD or FOG−/FOG+) effect sizes of the APA parameters were examined by calculating the Cohen’s d value and was classified according to its absolute value as small (0.20–0.49), moderate (0.50–0.79) or large (≥0.80) [36]. 

Linear correlations were used to assess the association between GI metrics, whereas the concurrent validity of the GI metrics with clinical scales was assessed using Spearman’s correlation coefficient, as the latter were not normally distributed. To interpret the magnitude of the correlation coefficients, the following guidelines from [37] were followed: for absolute values between 0 and 0.19, a very slight relationship; between 0.20 and 0.39, a slight one; between 0.40 and 0.59, moderate relationship; between 0.60 and 0.79, a strong one; and between 0.80 and 1, very strong.

The method sensitivity was assessed by comparison of the GI metrics through unpaired *t*-test between PD and ELD, and between PD without and with freezing of gait (FOG− and FOG+). The experience of FOG was determined by the positive response to the item 1 of the Characterizing Freezing of Gait Questionnaire (C-FOG). 

The *p*-value for statistical significance was set at 0.05.

## 3. Results

### 3.1. Participants’ Demographics and Clinical Assessment

The PD and ELD groups did not differ in age, weight, and height (PD 17 Males, mean (SD), age [yrs]: 73.9 (6.2), body mass [kg]: 68.4 (10.8), body height [cm]: 169.0 (6.0)); ELD 5 Males, mean (SD), age [yrs]: 68.9 (7.5), body mass [kg]: 70.6 (13.0), body height [cm]: 169.1 (6.8), *p* = 0.07; *p* = 0.63, *p* = 0.97). The clinical features of the PD group are summarized in Table 2.

### 3.2. Validation of Body-Fixed Sensor Gait Initiation Metrics

A good agreement of the IMUs compared to the reference system was observed for the APA timing measures through both the Bland–Altman method and linear correlation (Figure 2B–F). There was no bias in the measurement methods since the bias was very close to 0 in all variables. In line with the results of Bland–Altman analysis, the ICC analysis also showed a satisfying agreement between the measurements made using the two systems. In fact, all the parameters revealed an excellent agreement (Table 3).

### 3.3. MAEs Values

No significant differences were found between PD and ELD subjects considering MAEs for APA onset (mean ± SD [s], PD 0.00 ± 0.02, ELD 0.01 ± 0.01, *p* = 0.50), toe-off ([s], PD −0.01 ± 0.02, ELD −0.01 ± 0.02, *p* = 0.92) and heel-strike ([s], PD −0.01 ± 0.01, ELD −0.01 ± 0.01, *p* = 0.39) instants. Moreover, for the MAEs values of the parameters APA and Swing durations, no significant differences were found ([s] PD −0.02 ± 0.03, ELD −0.02 ± 0.03, *p* = 0.68; PD 0.00 ± 0.02, ELD 0.01 ± 0.02, *p* = 0.66, respectively).

### 3.4. Correlations between Gait Initiation Metrics

Significant positive correlations were found by examining the association among the GI metrics in persons with PD, as depicted in Figure 3. Time-to-APA and APA duration were correlated with Time-to-toe-off and Time-to-heel-strike, through strong and very strong associations, respectively. Furthermore, a strong correlation between Time-to-toe-off and Time-to-heel-strike was found, whereas the swing phase parameter was slightly correlated with Time-to-heel-strike.

### 3.5. Correlation between Gait Iniation Metrics and Clinical Scales

Figure 3 reports the correlation analysis between clinical outcomes and APAs parameters. GI metrics showed statistically significant correlations with the clinical outcomes. In particular, APAs parameters showed moderate-to-strong correlations with the assessment of episodic symptom FOG in persons with PD (Figure 3). Three parameters, Time-to-toe-off, Time-to-heel-strike, and APA duration, correlated negatively with the MDS-UPDRS item 3.11 (i.e., FOG Figure 3A) and the C-FOG questionnaire (Figure 3B). All the GI metrics, except for the swing phase duration, showed moderate positive correlations with the dynamic stability (FSST, Figure 3B). In addition, the time-to-APA correlated negatively with the score of the lower extremity function (i.e., SPPB and its item 3, Figure 3B) and the executive functions (FAB and its sub-item planning, Figure 3B). Time-to-toe-off, Time-to-heel-strike, and APA duration correlated positively with the assessment of the body rigidity (i.e., MDS-UPDRS Rigidity, Figure 3A).

### 3.6. Differences in APAs Timing Parameters between PD and ELD

PD showed a significantly longer time phase of APA with respect to that of ELD. Furthermore, toe-off and heel strike events were significantly delayed from stimulus release in PDs compared with healthy subjects (Table 4). On the other hand, the Time-to-APA and the swing phase duration metrics were comparable between the two groups, PD and ELD.

When comparing FOG− and FOG+, it emerged that FOG+ showed a reduction of APA duration with respect to FOG− (Table 4).

## 4. Discussion

In this study, we investigated the validity, accuracy, and sensitivity of a single IMU-based method for cued GI assessment in persons with PD.

Our findings confirmed the validity of the algorithm for the population under investigation. Results demonstrated low errors and good agreement between the single IMU-based system and the force platforms acquisition system for the estimation of the APAs timings. The APAs movements are needed to prepare the body for the upcoming disturbance, they are the first line of defense that the central nervous system (CNS) uses to maintain and restore balance when the equilibrium state is perturbed or changed [38]. Human vertical posture is inherently unstable due to the high location of the center of mass (COM), small base of support area, and multiple joints between the feet and the body’s COM position [39,40]. When a standing person performs a quick movement, as the gait initiation, the mechanical coupling of body segments leads to postural perturbations that may endanger fragile balance [41]. By activating the trunk and leg muscles, prior to a forthcoming predictable body perturbation (i.e., APAs), CNS minimizes the risk of losing equilibrium. It is well established that these postural adjustments are altered in subjects with PD [6]. However, so far, home-based exercises using IMUs-based system are focused on steady-state gait, therefore, not including the GI phase. Our findings encourage the implementation of exercises for APA training with out-of-lab system in persons with PD, as events such as the APA onset, the toe off, and the heel strike of the swing leg are detectable accurately and effectively from a single IMU positioned on the lower back. This sets the stage for future developments to improve the home-based training in PD. In fact, the significant correlations between parameters from IMU and force platforms support the possibility to adopt a single IMU to assess APAs outside a laboratory setting. 

Concerning the sensitivity, GI metrics extracted from a single IMU were able to discriminate between PD and elderly on the basis of APAs alterations. More precisely, persons with PD showed prolonged APAs compared to those of elders during GI (Table 4, APA Duration parameter). This finding was consistent with the ones published by previous studies [2,42,43] that have used the gold standard system to investigate the differences in the APAs timings between persons with PD and elderly subjects. Halliday et al. [42] found preserved temporal and spatial patterns of GI in elderly and PD subjects, but also increased APA phases durations in PD patients compared to those of the elderly, which, in turn, were longer than those of the young subjects. This trend suggests a progressive slowing of GI patterns from the young to the elderly, up to the PD subjects. Moreover, Crenna et al. [2] and Burleigh-Jacobs et al. found [43] that the APA duration was prolonged in PD subjects and that could be improved using external stimuli. The same prolonged duration has been detected also by the method here presented, supporting the validity of a single IMU-based system to investigate alterations in the APAs pattern. 

Furthermore, the algorithm has also detected differences within the PD group, between FOG- and FOG+. The alterations of APAs in PD are the result of the pathological characteristics of the disease that compromises not only the movement execution, but also movement preparation involving postural features related to the dopaminergic deficit [44]. In fact, interesting correlations between APA duration and clinical features emerged. The timing alteration of APAs was related to worst dynamic stability and greater limbs rigidity, assessed by FSST and MDS-UPDRS—Rigidity sub-score, respectively. Limb rigidity and dynamic stability were strongly connected, as body rigidity decreased the ability to control the center of mass with the feet in place, requiring frequent steps to maintain balance [45]. Consequently, it is not surprising that these characteristics were significantly correlated with the APAs duration. Instead, the strong negative correlation between APAs timing alteration and the FOG symptom was unexpected. Longer durations of the APA phase (i.e., worst performance) was associated to a lower severity of FOG. This association reflects the differences emerged between FOG+ and FOG−, with FOG+ showing a reduction of APAs duration compared to FOG-, indicating a better performance of FOG+. This result was opposite to what was expected, since FOG symptom is associated with less automatic gait and more impaired postural transitions compared to persons with PD who do not have FOG [46]. This result could be ascribed to the use of the auditory stimulus (i.e., “go”) prior to the step, as the evidence regarding the external cueing effects in patients with PD with FOG suggest that cue-trigger use improves gait parameters [47,48], especially for the preparation of GI [49,50]. Indeed, it has been shown that auditory cues appear to make use of a prompt motor entrainment to an external beat, activating the frontoparietal control and motor cerebellar networks to override internal rhythm deficits of the basal ganglia [51]. Therefore, we could hypothesize that FOG+ participants had greater benefits from the auditory cues with respect to FOG− participants. It is interesting noticing that the APA parameters strongly correlated with the two sub-scores of the MDS-UPDRS III associated to limb rigidity and FOG, but not with its overall score. This could be related to the fact that the final score of the MDS-UPDRS III is obtained by the sum of motor items that are not only specific to balance stability and gait, but also related to several other symptoms (e.g., speech, facial mimic, finger taps), which are probably not relevant for the APA performances.

The APA Onset, quantified by the parameter Time-to-APA, was comparable between person with PD and healthy subjects. This was probably due to the stimulus used. In fact, sensory cues, such as the auditory ones, can facilitate GI in PD patients, speeding up the APA reaction onset [52]. APA onset moderately correlated with the SPPB, FAB and its sub-part specific for planning. Greater latencies in the onset of APAs were associated with less dynamic stability and greater impairment of executive functions, confirming that the preparation of GI is associated with both the motor and cognitive domains. It should be noted that, among GI metrics, APA onset parameter was the measure that most reflected motor-planning processes, being associated with the FAB-2 planning score that assess the residual functions of the frontal lobe structures mainly delegated to the movement control. This correlation further highlighted that the parameters extracted from the IMUs are related to pathophysiological mechanisms. 

All of the above correlations between the GI metrics and the clinical evaluations of persons with PD demonstrated that the concurrent validity of the single IMU-base algorithm was good.

The present study has some limitations. First, the algorithm provides only temporal measures of gait initiation and not information regarding the COP displacement or the APA magnitude. Hence, the algorithm should be improved by integrating these additional measures. Second, we tested the accelerometer-based method for the quantification of the APA only during gait initiation. Future studies should test the APA detection algorithm on the performance of other daily tasks. Finally, the PD participants were classified between 1 and 3 according to the H&Y scale (mild to moderate PD). Thus, the ability of the algorithm to detect differences in more homogeneous groups should be further investigated.

## 5. Conclusions

The acceleration -based algorithm tested in this study is a promising tool for assessing GI of persons with Parkinson’s Disease. Results of the validity procedure conducted in this study demonstrated a strong agreement between a single IMU-based measurement system and the platforms acquisition system to estimate the APAs timings. In addition, the GI metrics were associated with clinical features of persons with PD, being able to discriminate between persons with PD who experienced freezing of gait and those that did not. APA impairment is a common feature among subjects with different neurological diseases and, therefore, future studies to validate the method on subjects affected by other neurological disorders, such as Multiple Sclerosis, are encouraged. In fact, a recent study highlighted that in the early stage of multiple sclerosis, although the APA of GI are strongly affected, there are no significant alterations of the executive phase [53], suggesting that APA is probably a reliable biomarker for early detection of motor deficits. Our results support the development of monitoring and telerehabilitation protocols through the use of a simple GI testing system. In fact, to obtain long-term effects of gait rehabilitation, it is necessary to train the APAs, which are needed to walk as physiologically as possible. Here we have shown that it is possible to detect the fundamental GI events (i.e., APA onset, toe-off and heel strike of the swing leg) starting from a single wearable sensor in PD. This sets the basis for the development of home-based training including exercises on the preparation phase, never realized so far. The GI testing system here presented could be a valuable tool for home-based rehabilitation programs yielding to a high patient adherence and a low risk of adverse events. Future studies should indeed investigate the patient satisfaction with the usability of the system, as well as the potential improvement of walking performances and quality of life aspects related to the use of the single sensor system for GI detection in persons with neurological diseases. 

## Figures and Tables

**Figure 1 sensors-22-02668-f001:**
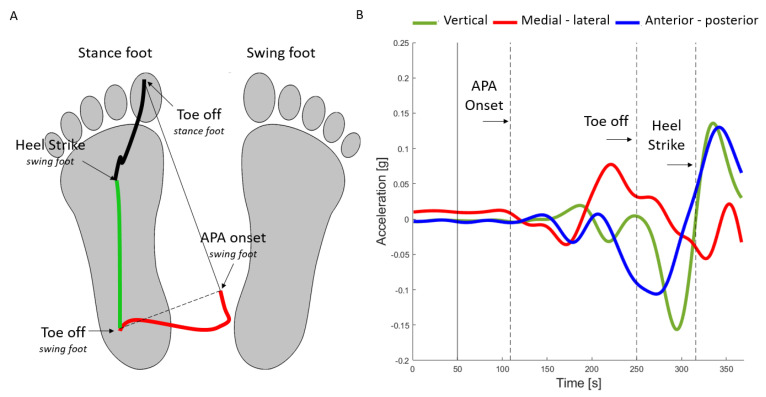
(**A**) COP displacement and (**B**) Acceleration signal during the gait initiation prior to the step in a healthy subject. In the panel A, the COP trace from APA onset to APA end (i.e., toe off) is reported in red and during the swing phase in green. In the panel B the first vertical line represents the auditory cue release and the next 3 vertical lines are the points derived by the algorithm, APA onset, swing leg toe-off and heel strike.

**Figure 2 sensors-22-02668-f002:**
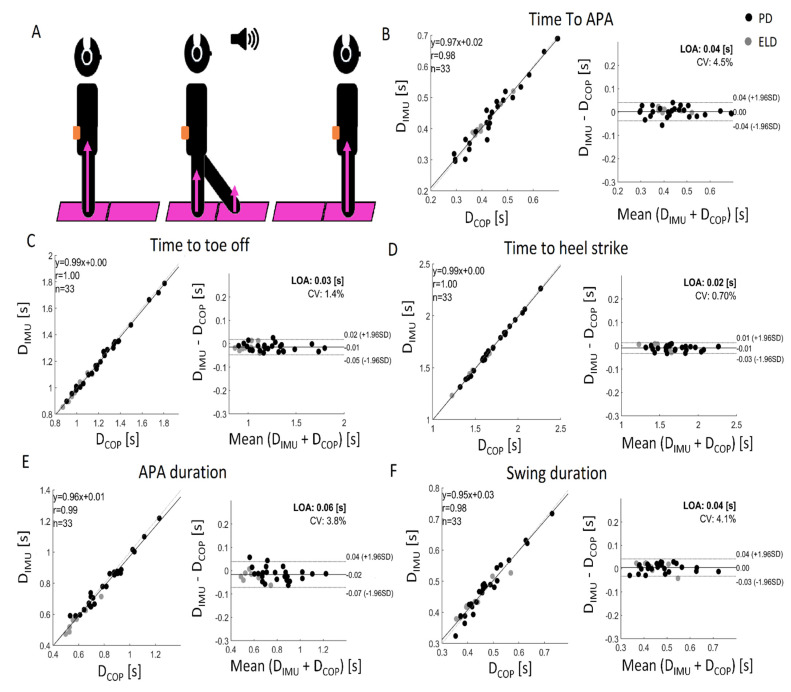
A schematic representation of the experimental paradigm, where the subjects stood on the first platform and took a step forward on the second platform when the stimulus arrived in the headphone (**A**). Bland–Altman plots of APA measures are reported from (**B**–**F**). PD persons with Parkinson’s disease; ELD elderly.

**Figure 3 sensors-22-02668-f003:**
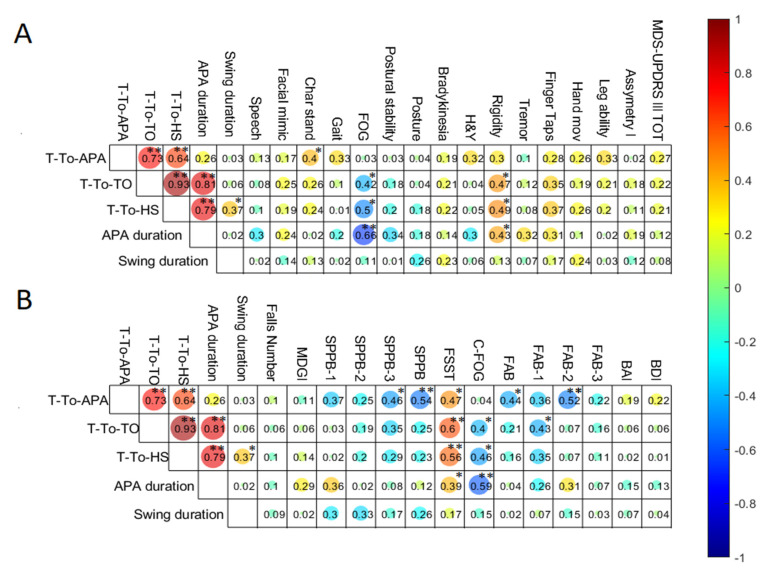
Correlations between gait initiation metrics and clinical outcomes for MDS-UPDRS III and its sub-items (**A**) and the other clinical scales (**B**). Absolute values are reported in the boxes, whereas the circle color indicates the positive (red) or negative (blue) degree. Significant correlation indices are indicated with * *p* < 0.05 and ** *p* < 0.01. H&Y stage, Hoehn and Yahr stage; MDS-UPDRS III, Unified Parkinson Disease Rating Scale Part III; MDGI, modified Dynamic Gait Index; SPPB short physical performance battery; FSST Four Step Square Test; C-FOG, Characterizing Freezing of Gait; FAB Frontal Assessment Battery; BAI, Beck Anxiety Inventory; BDI-II Beck Depression Inventory.

**Table 1 sensors-22-02668-t001:** Schematic summary of the main steps of the algorithm for the APA instants identification.

Parameter	Axis	Temporal Window [s]	Description
Identification of anticipatory postural adjustments onset			
APA_change_	AP, ML, V	[0.5 − t_s_: t_s_ + 1.2]	Time point where significant change of mean signal occurs
APA_onset_	AP, ML, V	[t(APA_change_) − t_s_: t(APA_change_)]	From t_APA__change_ search backwards for the time point of the first local minimum
Identification of toe-off of the swing limb			
Peak_va_max_	V	[0.8 + t_s_: t_s_ + 2]	Time point of the signal positive peak over a predefined threshold
I_Peak_FS_	V	[t (Peak_va_max_) − 1.5: t (Peak_va_max_) − 0.5]	Time point of the first step calculated at the peak closest to the temporal window closure
TO_swl_	V	[t (Peak_va_max_) − 1.5: t (I_Peak_FS_) ]	The zero-crossing time point just before the I_Peak_FS_
Identification of heel strike of the swing limb			
HS_swl_	V	[t(TO_swl_): t(I_Peak_FS_)]	Time point where the signal crossed the 20% value of the vertical accelation signal at I_Peak_FS_

t_s_: time of stimulus; APA_onset_: anticipatory postural adjustments onset; TO_swl_: toe-off of the swing limb; HSswl: heel strike of the swing limb; AP: anterior–posterior axis; ML: medio-lateral axis; vertical axis.

**Table 2 sensors-22-02668-t002:** Clinical characteristics of the sample of persons with Parkinson’s disease enrolled in this study.

	Median	(1st–3rd Quartile)
Number of falls	0	(2.0–2.0)
H&Y	3.0	(2.5–3.0)
MDS-UPDRS III	42.5	(35.5–54.0)
MDGI *	50.0	(42.0–55.0)
SPPB	9.0	(2.0–4.0)
FSST	13.5	(11.2–20.2)
C-FOG	25.0	(0.0–47.0)
FAB	16.0	(14.0–18.0)
BAI	17.0	(8.0–31.0)
BDI-II	13.0	(10.0–16.0)

* H&Y stage, Hoehn and Yahr stage; MDS-UPDRS, Unified Parkinson Disease Rating Scale Part III; MDGI, modified Dynamic Gait Index; SPPB short physical performance battery; FSST Four Step Square Test; C-FOG, Characterizing Freezing of Gait; FAB Frontal Assessment Battery; BAI, Beck Anxiety Inventory; BDI-II Beck Depression Inventory.

**Table 3 sensors-22-02668-t003:** Intraclass correlation coefficients (ICC) of gait initiation (GI) metrics extracted from the inertial measurement unit and the force platforms.

GI Metrics	ICC	Lower Bound	Upper Bound	Cronbach’s Alpha
Time-to-APA	0.99	0.98	0.99	0.91
Time-to-toe-off	0.99	0.98	0.99	0.99
Time-to-heel-strike	0.99	0.99	1.00	1.00
APA duration	0.99	0.97	0.99	0.99
Swing duration	0.98	0.97	0.99	0.98

**Table 4 sensors-22-02668-t004:** Gait initiation metrics for both groups, PD and ELD.

Parameters	PD	FOG−	FOG+	ELD	Cohen’ d PD/ELD FOG+/FOG−	
Time-To-APA [s]	0.45(0.12)	0.44(0.10)	0.46(0.13)	0.42(0.05)	0.30	0.18
Time-To-Toe-Off [s]	1.26(0.24)	1.31(0.20)	1.21(0.26)	1.00(0.09) *	1.07	0.45
Time-To-Heel-Strike [s]	1.73(0.26)	1.80(0.23)	1.66(0.27)	1.44(0.15) *	1.10	0.55
APA duration [s]	0.81(0.17)	0.88(0.12)	0.75(0.19) **^+^**	0.59(0.03) *	1.20	0.78
Swing phase duration [s]	0.47(0.09)	0.49(0.12)	0.46(0.06)	0.43(0.07)	0.48	0.38

* *p* < 0.05 PD vs. ELD; **^+^** *p* < 0.05 FOG− vs. FOG+.

## Data Availability

The dataset used and/or analyzed during the current study are available from the corresponding author on reasonable request.

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
