# Peer review of "Events Detection of Anticipatory Postural Adjustments through a Wearable Accelerometer Sensor Is Comparable to That Measured by the Force Platform in Subjects with Parkinson’s Disease"

_sensors, 2022, doi:10.3390/s22072668_

Round 1
Reviewer 1 Report
Thanks for the opportunity to review the manuscript titled, " Events detection of anticipatory postural adjustments through a wearable accelerometer sensor is comparable to that measured by the force platform in subjects with Parkinson's disease". The manuscript aims to evaluate the validity of using 1 IMU to detect the gait initiation measures in people with Parkinson's disease.
Results of the study showed that the gait initiation measures identified from accelerometric data using a previously published algorithm demonstrated a good level of agreement with that captured by force plates. Moreover, the IMU-based gait initiation measures showed a significant correlation with several clinical scales and were able to differentiate the PD patients and healthy elderly.
The study supports the use of a single IMU to detect the gait initiation and several gait events in PD patients. The authors reviewed the relevant article and identified the shortcomings of studies that used IMU-based sensors to detect the APA. I think the study provides scientific evidence to guide the development of wearable equipment that can be used to monitor the disease progression in PD patients.
One of my concerns are
- In the introduction, please elaborate on the impairment of APA in PD patients. What impairments have been reported in previous studies? (Line52 – 53)
- I have difficulty understanding the algorithm that locates the temporal instants from the accelerometer without referring back to the original reference [3]. I hope that the authors can revise the descriptions, and perhaps the authors could use tables/figures to facilitate the illustration.
- In the discussion section, the authors suggested that the APAs timing alteration were associated with the MDS-UPDRS. In fact, the APAs measures are associated with a few items in the MDS-UPDRS only. The authors may need to modify this statement and suggest why there is no correlation between the total score of MDS-UPDRS and APAs measures.
Other suggestions/Comments
- Although the authors have defined Gait initiation as GI, I still see the authors used the full term in the manuscript frequently. Please also check whether this happens for other abbreviations.
- Please standardise 'force plate' and 'force platform'.
- Please justify the sample size in the ‘participants’ section.
- Line 105 – 119. There are no references provided for most of the clinical assessments.
- Line 131: Can you provide references/rationale for selecting the filtering frequency of 0.2 – 4.5Hz?
- Line184: what is the meaning of 'end of APA wave form'? Can you locate it in figure 1B?
- I believe that the equation in Figure 2A should be y = 0.99x +0.00.
- Please provide effect size estimates in table 3.
- Line 314: What are the major findings from reference [2,33,34].
- What is the number of PD patients belonging to FOG- and FOG+ groups?
Author Response
Reviewer #1
References to pages (P) and lines (L) are related to the new version attached of the manuscript with major changes highlighted in red
Thanks for the opportunity to review the manuscript titled, " Events detection of anticipatory postural adjustments through a wearable accelerometer sensor is comparable to that measured by the force platform in subjects with Parkinson's disease". The manuscript aims to evaluate the validity of using 1 IMU to detect the gait initiation measures in people with Parkinson's disease.
Results of the study showed that the gait initiation measures identified from accelerometric data using a previously published algorithm demonstrated a good level of agreement with that captured by force plates. Moreover, the IMU-based gait initiation measures showed a significant correlation with several clinical scales and were able to differentiate the PD patients and healthy elderly.
The study supports the use of a single IMU to detect the gait initiation and several gait events in PD patients. The authors reviewed the relevant article and identified the shortcomings of studies that used IMU-based sensors to detect the APA. I think the study provides scientific evidence to guide the development of wearable equipment that can be used to monitor the disease progression in PD patients.
We thank the reviewer for appreciation and careful reading. The text has been further improved in accordance with his/her indications.
One of my concerns are
- In the introduction, please elaborate on the impairment of APA in PD patients. What impairments have been reported in previous studies? (Line52 – 53)
We apologize for not having explicitly introduced the main impairments. We have now added a concise description of the APA alterations in PD. (P2 L54-58)
- I have difficulty understanding the algorithm that locates the temporal instants from the accelerometer without referring back to the original reference [3]. I hope that the authors can revise the descriptions, and perhaps the authors could use tables/figures to facilitate the illustration.
We have modified the text and added a table (i.e. Table 1) to facilitate the reading. (P5)
- In the discussion section, the authors suggested that the APAs timing alteration were associated with the MDS-UPDRS. In fact, the APAs measures are associated with a few items in the MDS-UPDRS only. The authors may need to modify this statement and suggest why there is no correlation between the total score of MDS-UPDRS and APAs measures.
It was a refuse, now the statement has been modified referring to the subscore rigidity of the of MDS-UPDRS III (MDS-UPDRS III – Rigidity subscore P10 L368). The lack of correlation between MDS-UPDRS III and APAs measures could be ascribed to the fact that MDS-UPDRS-III scale contains not only subscores specific to balance stability and gait but also to several other symptoms (e.g. speech, facial mimic, finger taps) which are probably not relevant for the APA performance. This clarification has now been added as requested P 11 L 386-392.
Other suggestions/Comments
- Although the authors have defined Gait initiation as GI, I still see the authors used the full term in the manuscript frequently. Please also check whether this happens for other abbreviations.
We have standardized the text with the acronym GI, and checked for the other terms.
- Please standardise 'force plate' and 'force platform'.
Done.
- Please justify the sample size in the ‘participants’ section.
As requested we have justified the sample size in the ‘Participants’ section P3 L112-118
- Line 105 – 119. There are no references provided for most of the clinical assessments.
The references have been provided in the manuscript.
- Line 131: Can you provide references/rationale for selecting the filtering frequency of 0.2 – 4.5Hz?
We used the same pre-processing filtering of the original study reported online as supplementary material of Gazit et al [3]. However, this is in line with other studies, since the COP movement occurs slowly and consequently the reference frequency of the signal is in the low band.
- Line184: what is the meaning of 'end of APA wave form'? Can you locate it in figure 1B?
The end of the APA waveform is determined by the toe-off time point. In the figure the second dotted line identifies this point. Now the text (P 6 L215-216) and the figure 1B have been modified to make this clearer.
- I believe that the equation in Figure 2A should be y = 0.99x +0.00.
Thank you for pointing this out, we have corrected the figure.
- Please provide effect size estimates in table 3.
As requested, we have calculated the between-group effect size (P6 L231-234) and added the values in table 4 (# updated, we have added a table see above point 2 major conserns).
- Line 314: What are the major findings from reference [2,33,34].
The main findings of the cited literature have been reported more extensively in the text as requested. Our results have shown a prolonged APA duration of PD with respect to elderly and this pattern is similar to that reported in previous studies that have used the force platform (i.e. gold standard system) as measurement system (P10 L353-361).
- What is the number of PD patients belonging to FOG- and FOG+ groups?
We apologize for not reporting this information, we added it in the revised paper (P3 L101-102 i.e. 12 FOG- and 13FOG+).

Reviewer 2 Report
1, Please improve the quality of figures. Some of them are barely readable.
2, It was very interesting to see how the sensors were made, details including processing method, materials (using nano materials or not?), sensing principle should present in the manuscript.
3, More abundant details of analysis methods should be used to verify the quality of the GI testing system.
4, Some more up to date literature can be cited.
5, The language needs to be refined.

Author Response
Reviewer # 2
References to pages (P) and lines (L) are related to the new version attached of the manuscript with major changes highlighted in red
In the manuscript entitled “Events detection of anticipatory postural adjustments through a wearable accelerometer sensor is comparable to that measured by the force platform in subjects with Parkinson’s disease.”, the authors propose a novel wearable accelerometer sensor, which creatively focuses on gait initiation (GI) instead of steadystate gait. Compared with the gold-standard system (the force platform), the GI testing system pays more attention on monitor Anticipatory Postural Adjustments (APAs), and the potential validity of the GI testing system improve APA rehabilitation treatment in Parkinson’s disease (PD) patients. Firstly, a wearable accelerometer sensor is used to continuously detect the functional GI events. Then, the temporal parameters extracted from the accelerometric data show a strong agreement with the parameters extracted from force plates. Finally, twenty-five PD and eight elderly subjects are participated in a simple GI testing experiment. The concept and demonstrations were also introduced to show its novelty and superiority. Considering this work is of great scientific and practical application potential, I would like to recommend the publication of this manuscript in Sensors after the following issues are addressed.
We thank the reviewer for appreciation and careful reading. The text has been further improved in accordance with his/her indications.
1, Please improve the quality of figures. Some of them are barely readable.
We have revised the figures as requested, and now they should be more readable.
2, It was very interesting to see how the sensors were made, details including processing method, materials (using nano materials or not?), sensing principle should present in the manuscript.
We have reported in the text all the available information about the commercial CE-marked IMU measurement system used (P3 L141-148)
3, More abundant details of analysis methods should be used to verify the quality of the GI testing system.
We did not fully understand the request. We have used three methods to assess the agreement between the two systems of measurement, Bland-Altman, MAE, and ICC (P6 L220-229). We reported this information in the Methods section.
4, Some more up to date literatures can be cited.
We have updated the literature as requested (e.g. [8], [10], [50], [53])
5, The language needs to be refined.
The language of the entire manuscript has been reviewed and edited by an English speaker.

Reviewer 3 Report
The paper is interesting, clear, pleasant to read and written in a good English. The rationale of the work conducted is clear and reasonable, and the methodology followed during the experimental part is sound. The results are clear, well presented, and fairly discussed, and the conclusions are reasonable, too.
I just have a couple very minor points to be addressed before the acceptance of the manuscript, as indicated below:
- Which software did you employ for the statistical analysis?
- I would stress a bit more about the study limitations
- In the Conclusions, I would add some more tips about the future developments of this work, given the fairly good results obtained
Author Response
Reviewer # 3
References to pages (P) and lines (L) are related to the new version attached of the manuscript with major changes highlighted in red
The paper is interesting, clear, pleasant to read and written in a good English. The rationale of the work conducted is clear and reasonable, and the methodology followed during the experimental part is sound. The results are clear, well presented, and fairly discussed, and the conclusions are reasonable, too.
We are grateful to the reviewer for understanding and appreciating the work.
I just have a couple very minor points to be addressed before the acceptance of the manuscript, as indicated below:
- Which software did you employ for the statistical analysis?
Data processing and Statistical analyses were performed using Matlab and SPSS. This information has been added in the statistical analyses paragraph P6 L219-220.
- I would stress a bit more about the study limitations
We understand and agree with the reviewer's observation. We have included a paragraph related to the limitations of the study P11 L408-416.
- In the Conclusions, I would add some more tips about the future developments of this work, given the fairly good results obtained
thanks for the suggestions, we have introduced potential future developments (P 12 L424-430, L 436-441).

Round 2
Reviewer 1 Report
The authors have addressed my comments and I am satisfied with the quality of the revised manuscript.
This manuscript is a resubmission of an earlier submission. The following is a list of the peer review reports and author responses from that submission.
Round 1
Reviewer 1 Report
General Comments:
The methodology is sound and the result is prepared neatly. The construct of the paper overall is quite good. It seems that a significant amount of time and effort has been put into preparing this paper. However, there are some concerns regarding the content to which this manuscript is prepared.
- Figure 1 should be revised.
- Details of the methodology
There are quite a substantial number of papers using single IMUs to detect gait events, some even estimating posture and joint angles. How does this work stand out from the rest? More emphasis should be placed on how the system was modified or adjusted to detect gait events of PD patients. I believe this will make it a much stronger paper.
Specific comments are below:
Page3 line 114 Procedures and throughout the manuscript
Though the authors mention that the gait event detection algorithm used was referenced from Gazit et al. it would be nice if the authors could mention in brief how the data collected by the IMUs were processed here.
“Data were recorded at a sample rate of 142 Hz from an IMU”
Please specify exactly what kind of data was collected. I believe that this is acceleration measurements but the term IMUs is very broad and could imply other data such as angular velocity, geomagnetic measurements, etc.
In addition, why was this sampling rate chosen?
The authors used:
- Audio feedback
- Acceleration data
- Force plate data
Simultaneously for their experiments. Can you add details into how all these data were synchronized taking into account the various sampling rates of different systems? This would greatly affect the accuracy of the collected data and the results.
Page5 line 179 Fig 1
Fig 1 A: Seems to be elongated, please check the aspect ratio of the figure.
Fig 1 B-F Too small to decipher the contents. Suggest enlarging. What is the difference between the grey and black dots?